# Universal peptide synthesis via solid-phase methods fused with chemputation

Jacopo Zero [1,2], Tristan J. Tyler [1,2] & Leroy Cronin [1] ✉

Since the advent of automated solid-phase peptide synthesis (SPPS), many commercial platforms have been developed, facilitating cutting-edge research across many biochemical fields. However, despite considerable technological advancements, these systems remain limited in flexibility and chemical capability. Herein, we present a fully automated programmable platform that combines the efficiency of SPPS with the chemical flexibility of a Chemical Processing Unit (Chemputer). SPPS protocols, from resin swelling to peptide precipitation, are captured and automated using the Chemical Description Language (χDL), affording peptide sequences in high purity (>79%) and on a multi-milligram scale. Owing to the modularity of the platform, valuable transformations are integrated into the workflow, including ring-closing metathesis, copper-catalyzed azide-alkyne cycloaddition, and native chemical ligation. These tailored modifications are carried out in one, uninterrupted synthetic protocol, performing up to 1635-unit operations, executed over 85 h of activity, producing peptides such as GHRH(1-29), Semaglutide, and Capitellacin, finally unlocking bottlenecks in automated SPPS.

With continuous technological advancements in the field of robotics, automation is set to play an increasingly vital role in chemical, biological, and material sciences[1]. By automating both repetitive and complex tasks, researchers can drastically enhance the efficiency, scalability, and safety of synthetic processes[2–5]. This step-change accelerates the discovery of new compounds whilst minimizing human error, ensuring consistency, and improving reproducibility[6]. Moreover, automation liberates researchers to focus on more creative, strategic tasks, fueling design and innovation. Despite these technological advances, the systemic adoption of automated chemical synthesizers remains stagnant, with the majority of chemical synthesis still performed manually. The reluctance to integrate automation predominantly stems from a lack of universality, with current automated technologies tending to occupy highly specialized research areas, for example, the automated synthesis of oligopeptides[7], oligonucleotides[8], oligosaccharides[9], and small molecules via MIDA boronate assembly[10]. Whilst these systems boast incredible feats, they remain limited in their broader utility.

Peptide synthesis is a prototypical example of this conundrum. Having been one of the first fields to successfully integrate automation in 1965[7], solid-phase peptide synthesis (SPPS) has undergone a remarkable transformation, driven by the continual development of commercially available automated platforms. These innovations have revolutionized the biochemical field, enabling the rapid and efficient assembly of peptide sequences with precise control over chemical functionality and structural diversity. However, manual intervention is still required for the final synthetic steps, such as peptide cleavage and ether precipitation. Additionally, due to the bespoke nature of these platforms, only a limited range of chemical operations can be integrated into the automation process[11].

The Chemical Processing Unit (Chemputer) was designed to overcome the inherent limitations of bespoke platforms and automate the multi-step synthesis of organic molecules under one, universal framework[12–14]. In this work, we integrate the precision and efficiency of solid-phase synthesis with the chemical versatility of the Chemputer to automate the synthesis of peptide substrates with subsequent chemical modifications (Fig. 1).

The programmable nature of the platform enables the complete automation of every synthetic step of SPPS, including resin swelling,

[1]School of Chemistry, University of Glasgow, University Avenue, Glasgow, UK. [2]These authors contributed equally: Jacopo Zero, Tristan J. Tyler.
✉e-mail: lee.cronin@glasgow.ac.uk

**Fig. 1 | General workflow for the fully automated synthesis of modified peptides on the Chemputer.** Reagents used during SPPS, and the accompanying procedures, are translated into a digital χDL script. A digital graph is then generated with the necessary modules to carry out the synthetic operations, and a physical platform is assembled accordingly. Finally, with the digital and physical requirements met, the platform can then execute the requisite synthetic operations, including SPPS, resin cleavage, ether precipitation, and subsequent desired chemical modifications, in a single automated procedure. Fmoc-AA$_n$-OH = 9-fluorenylmethoxycarbonyl protected amino acids; HATU hexafluorophosphate azabenzotriazole tetramethyl uranium, DIPEA *N*,*N*-diisopropylethylamine, TIPS triisopropylsilane, TFA trifluoroacetic acid, DMF *N*,*N*-dimethylformamide, DCM dichloromethane.

solid-phase peptide assembly, resin cleavage and sidechain deprotection, and ether precipitation. In stark contrast to commercially available peptide synthesizers, the modular construction of the Chemputer provides complete synthetic freedom, allowing for the facile introduction of various chemical operations, in complete automation, both during and after solid-phase peptide assembly. This advanced system carried out up to 1635-unit operations over 85 h of continuous activity, successfully generating traditionally challenging peptide sequences with high crude purities (>79%). The modular design of the Chemputer peptide synthesizer (SPPS Chemputer) allows for the seamless integration of various chemical modifications—including lipidation, ring-closing metathesis (RCM), bicyclization, copper-catalyzed azide-alkyne cycloaddition (CuAAC) click functionalization, cysteine arylation, directed oxidative folding, sidechain cyclization and derivatization, and native chemical ligation (NCL)—into a single uninterrupted automated protocol. Demonstrating that peptide synthesis, and in-situ chemical derivatization, can be fully programmed and executed on one platform.

## Results and discussion
### Programming peptide synthesis with χDL
As digital chemistry advances and automated synthesis becomes more widespread, standardizing how synthetic procedures and results are documented is crucial. The Chemical Description Language χDL[6,15,16], offers a way to digitally encode common synthetic operations in a format that is both human-readable and machine-executable. Chemical procedures are encoded within a χDL script, which lists the sequential chemical steps to be performed by a platform to synthesize

a target molecule. By emulating the workflow of the bench chemist, synthetic steps in χDL are represented as a sequence of physical processes such as Add, Purge, Filter, and more[14].

In addition to this, more complex and machine-oriented features have been developed, including blueprints, parallel execution, and real-time sensor monitoring, presenting an advantage that cannot be matched by humans[17]. For example, blueprints package long and complex synthetic operations into a concise single step. This is of particular interest for SPPS, which involves the iterative execution of several synthetic steps until the full-length peptide sequence is assembled. These individual stages of SPPS (resin swelling, Fmoc deprotection, amino acid coupling, resin cleavage, and ether precipitation) can be easily summarized into χDL blueprints, allowing for the generation of a concise and human-readable χDL digital synthetic script (Fig. 2).

Blueprints also offer a way to flexibly change the synthetic conditions via parameters. Utilizing parameters throughout the synthetic SPPS workflow enables amino acids, coupling reagents, reaction times, and repeat conditions to be readily exchanged or modified within the executable χDL file to meet synthetic requirements. In addition to this, blueprints for the chemical modification of peptides are organized in the same way. By translating literature procedures, chemical reactions typically performed manually by trained personnel can be digitally encoded and integrated into the automated SPPS workflow. Therefore, χDL successfully integrates complex and highly specialized chemical steps into the automated operations routinely executed on commercially available peptide synthesizers. Any procedure from the chemical literature can then be directly translated into a χDL script, fully

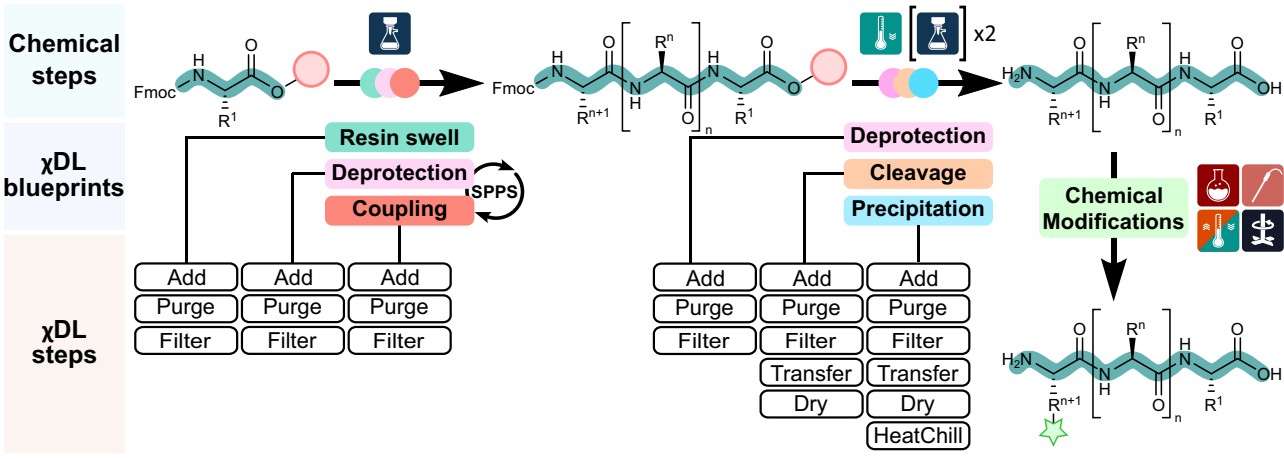

**Fig. 2 | Digital translation of the synthetic operations required for SPPS into χDL blueprints.** The synthetic chemical steps required to produce complex peptides are summarized in concise χDL digital blueprints, corresponding to the characteristic stages of SPPS. These, in turn, consist of a sequential list of χDL steps, encoding the chemical synthesis of a desired peptide target through specific physical operations to be performed by the automated platform.

encoding and digitizing not only SPPS but also the procedures required for post-assembly peptide modification.

## Chemputer hardware for the synthesis and modifications of peptides

Consisting of a series of pumps, valves, and reaction modules, the Chemputer is a modular programmable chemical synthesizer built to execute complex multi-step reactions[12–14]. A key feature of the Chemputer architecture is its discrete design, comprising of logically and physically self-contained units. These modules are interconnected through a liquid-handling backbone that links them to each other as well as to shared resources, such as reagents and solvents. The connectivity between modules and the backbone is represented via a digital graph, enabling the software to locate and utilize available hardware resources (Supplementary Fig. 3). This modular approach ensures the system is easily extendable, allowing individual modules to be modified or added without affecting the overall framework.

The base setup of the SPPS Chemputer (Fig. 3) comprised two fritted filters, one where peptide elongation and final resin cleavage take place (SPPS reactor), and a second jacketed filter, paired with a recirculating chiller, that is used during ether workup and peptide precipitation (Precipitating unit). Both the SPPS reactor and Precipitating unit are connected to the backbone via auxiliary valves, which provide pressurized nitrogen gas, via the pneumatic controller[14], and vacuum. The pneumatic controller consists of an array of solenoids that are used to deliver high active and low passive nitrogen flow, purging the resin solid support during the various stages of SPPS as required. Vacuum is fed through a diaphragm pump and is used to quickly evacuate the contents of the reactor into a pressure-resistant waste container. This ensures that the reactor is completely emptied between steps and the resin solid support is fully dried before cleavage.

## Fully automated synthesis of peptides

For the initial development of synthetic parameters, the challenging acyl carrier protein sequence fragment ACP(65-74) (**1**)[18] was chosen to benchmark the efficiency of the platform following standard Fmoc/tBu SPPS conditions. The optimal parameters were determined to be amide couplings consisting of 2 mL Fmoc protected amino acid (0.5 M in DMF), 2 mL coupling reagent hexafluorophosphate azabenzotriazole tetramethyl uronium (HATU) (0.45 M in DMF), and 0.5 mL *N,N*-diisopropylethylamine (DIPEA) (1 × 30 min). Fmoc deprotection consisted of two treatments of 20% piperidine in DMF (2 × 9 mL, 15 min total). Peptide cleavage from the resin solid support was performed

using a 10 mL cleavage cocktail of TFA/$H_2O$/TIPS (90:5:5, v/v/v), prepared before use in automation, with the individual components directly added to the SPPS reactor and mixed with low-pressure nitrogen flow for 2 h.

Following cleavage, the TFA mixture was transferred to the Precipitating unit containing pre-chilled diethyl ether (180 mL, −20 °C), added in parallelized automation during the cleavage step. The resin was washed with another 10 mL of TFA before being transferred to the Precipitating unit. The ether/cleavage mixture was then mixed, via nitrogen bubbling, for 30 min before the filtrate was transferred to a buffer flask. Finally, the peptide precipitant was washed three times with cold diethyl ether (3 × 30 mL, −20 °C), dried under vacuum, dissolved in MeCN/$H_2O$ (1:1, v/v), and transferred to the product flask awaiting analysis. To verify these conditions optimized for peptide **1**, the longer peptide sequences 18A (**2**), an amphipathic helical peptide[19], and GHRH(1-29) (**3**), the active component of growth hormone-releasing hormone[20] were then synthesized in full automation and analyzed by reversed-phase HPLC (RP-HPLC) and electrospray ionization mass spectrometry (ESI-MS) (Fig. 4). Despite the increased sequence length, all three peptides were synthesized with high crude purities (>87%) and in good yields (53 mg (41%), 195 mg (67%), and 261 mg (59%), respectively). Notably, when testing different reaction conditions, a high level of reproducibility was observed, illustrating the platform's reliability and consistency (Supplementary Table 1). Furthermore, to test the efficiency of the platform to incorporate sterically hindered N-alkyl amino acids, a N-methylated variant of 18A (N-Methyl-18A; **15**) was subsequently synthesized, incorporating two N-Methyl-Phe residues at positions 6 and 18 with high purity (79%) and good yield (62%) (see Supplementary Information 5.12 N-Methyl-18A (**15**)).

To investigate the capability of the platform to perform site-selective sidechain functionalization, the blockbuster therapeutic Semaglutide (**4**) (Brand name Ozempic®, Rybelsus®, and Wegovy®; Novo Nordisk) was then synthesized on the SPPS Chemputer (Fig. 5a). Prescribed in the treatment of type 2 diabetes and chronic weight management, Semaglutide was chemically derived from human glucagon-like peptide-1 (GLP-1)[21]. Designed to be a long-acting analogue of GLP-1, Semaglutide contains two unnatural functionalities, namely a 2-aminoisobutyric acid (Aib) group and a lysine acylated fatty acid moiety, consisting of two 8-amino-3,6-dioxaoctanoic acid (Ado) spacers, one γ-glutamic acid (Glu) linker, and a fatty C18 diacid (1,18-octadecanedioic acid)[22,23]. To that end, Fmoc-Aib-OH, Fmoc-Lys(Mtt)-OH, and Boc-His(Trt)-OH were introduced to the linear peptide sequence at the appropriate locations. Upon coupling of the final Boc-

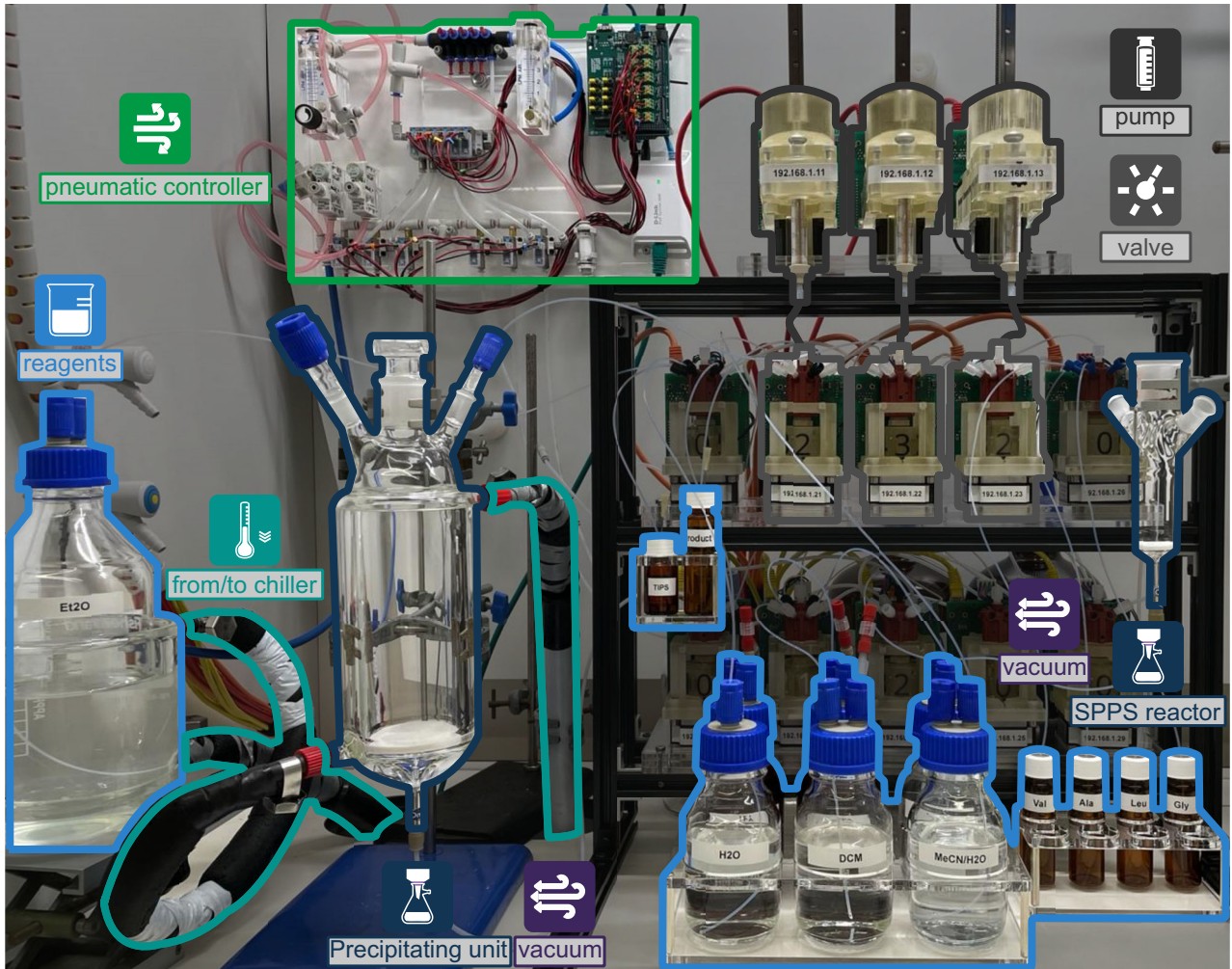

**Fig. 3 | Chemputer platform physical setup, including the base modules required for SPPS.** The modules used for the synthesis of linear peptides from commercially available starting materials are highlighted. The Chemputer liquid-handling backbone is used to transfer reagent solutions from one module to another. Two fritted filters (SPPS reactor and Precipitating unit) are used for peptide assembly, cleavage, and ether precipitation. The pneumatic controller is then used to feed nitrogen gas at two different flow rates to assist during the various stages of SPPS. A diaphragm pump (not shown) is connected to the filters' auxiliary valves through a vacuum-resistant waste container.

protected residue (His1), the Mtt protecting group of the Lys20 side-chain ε-amine was removed using a solution of HFIP/DCM/TIPS (70:25:5, v/v/v), prepared in automation (2 × 30 min). Then, the two Ado spacers, γ-Glu linker, and fatty diacid were sequentially coupled (2 × 1 h, 2 × 1 h, and 2 × 2 h, respectively) onto the free amine. Following these on-resin couplings, the peptide was cleaved, precipitated, and purified by RP-HPLC to afford 80 mg of Semaglutide (**4**; 18% yield, 71 steps).

Following standard synthetic practices, this process would have been considerably more time-consuming and labor-intensive, requiring multiple manual steps and potential intermediate purifications, not to mention the historically manual cleavage and precipitation. In contrast, use of the SPPS Chemputer platform enabled the streamlined and automated synthesis of Semaglutide, resulting in a high crude purity of 57% (Supplementary Fig. 9a), exceeding previously reported values for the total chemical synthesis of the peptide (36–39%)[24]. This highlights the capability of the Chemputer approach in accelerating complex peptide synthesis while improving both efficiency and product quality.

The next on-resin chemical modification chosen was peptide stapling, an effective strategy for stabilizing α-helices, important structural motifs that dictate the biological activity of various peptides and proteins[25]. All-hydrocarbon stapling, in particular, has emerged as a powerful method for stabilizing α-helices and has produced several examples of peptides with higher target affinity and with dramatically increased protease resistance[26]. To this end, NYAD-13 (**5**), a cell penetrating α-helical peptide shown to disrupt the formation of HIV-1 virus particles in vitro systems[27], was chosen to be the next target for automated synthesis (Fig. 5b). The linear sequence of NYAD-13 was first assembled on rink amide resin with the olefinic Fmoc-(S)-2-(4-pentenyl)Ala-OH (Fmoc-S5-OH) residues introduced at positions 4 and 8. Upon reaching the final amino acid (Ile1), the Fmoc group was not removed, and the RCM reaction was performed using 1st Generation Grubbs catalyst M102 (10 mM in degassed 1,2-dichloroethane, 2 × 2 h) sparged with argon. The peptide was then Fmoc deprotected, cleaved from the resin, ether precipitated, and purified by RP-HPLC to afford 138 mg of the all-hydrocarbon stapled peptide NYAD-13 (**5**; 63% yield, 32 steps).

Having demonstrated the capability of the platform to perform on-resin chemical modifications, the synthesis and subsequent in-solution chemical functionalization of peptides in a single automated workflow was then investigated. The first target was the automated synthesis of bicyclic peptides, synthetic molecules rotationally constrained into a rigid two-loop structure, termed peptide bicycles[28].

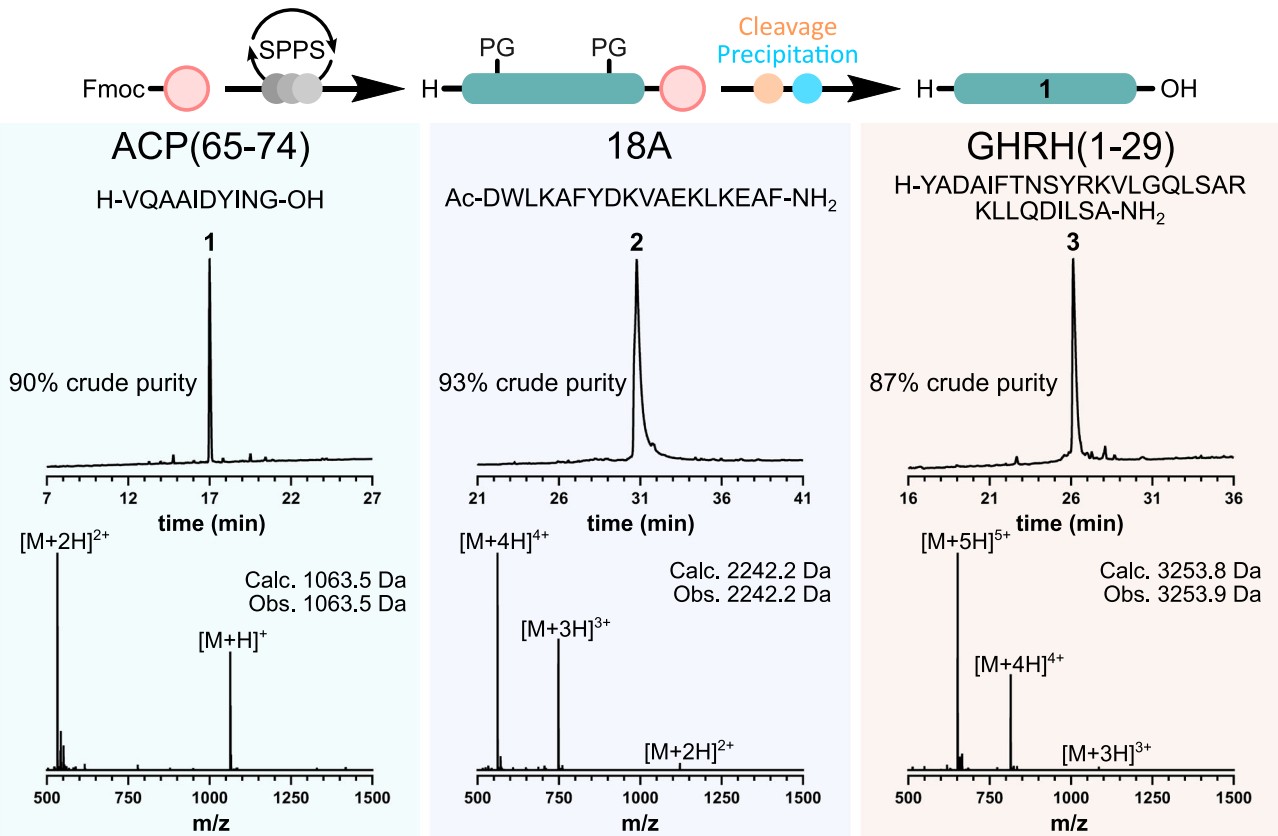

**Fig. 4 | Fully automated chemical synthesis of traditionally difficult peptide sequences.** Crude RP-HPLC traces (214 nm, 60 min 0–80% MeCN gradient) and corresponding ESI-MS of ACP(65–74) (**1**), 18A (**2**), and GHRH(1-29) (**3**). All peptides were synthesized in full automation, including assembly, cleavage, and ether workup. Relative peak areas were used to determine the efficiency of peptide assembly and were denoted as crude purity. Mass spectra were obtained from the summation of the time scale shown.

These constrained molecules are an emerging class of modified peptides that offer conformational rigidity, metabolic stability, and antibody-like affinity[29]. The chosen example was the peptide–bismuth bicyclic complex, generated by the interactions between three cysteine residues and a trivalent bismuth ion (Fig. 6a)[30]. To execute this chemical transformation, an additional reaction vessel and a magnetic hotplate stirrer were added to the base platform. The linear peptide was first assembled on rink amide resin, cleaved, and ether precipitated. The precipitated peptide was then dissolved in $H_2O$ before being transferred to the reaction vessel. Here, TRIS buffer (20 mM in $H_2O$), tris(2-carboxyethyl)phosphine (TCEP; 20 mM in $H_2O$), and $BiBr_3$ (60 mM in DMSO) were sequentially added. The reaction mixture was then stirred for 1 h before being purified by RP-HPLC, reporting 64 mg of the peptide–bismuth bicyclic complex (**6**; 33% yield, 29 steps).

The next automated chemical reaction targeted was the CuAAC click reaction. Click chemistry is an immensely powerful chemical technique that has found wide utility in the area of bioconjugation, a growing field that has been exploited in the study of cellular processes[31]. In this example, the reported cell-penetrating peptide penetratin was N-terminally equipped with an alkyne handle (4-pentynoic acid) to be functionalized at a later stage (Fig. 6b). Following automated assembly, cleavage, and ether precipitation, the alkyne containing peptide was then dissolved in DMF and transferred into a reactor preloaded with CuBr and the fluorescein azide (6-FAM-azide) and kept under a nitrogen atmosphere. The mixture was then further diluted to a peptide concentration of 0.5 mM and sparged with nitrogen (10 min). The reaction mixture was then stirred for 12 h and the crude peptide purified by RP-HPLC, affording 52 mg of the click

fluorescently labeled penetratin (**7**; 29% yield, 35 steps). Next, the SPPS Chemputer platform was used to automate the synthesis of a library of functionalized peptide targets via late-stage diversification. The chosen technique was cysteine arylation stapling using a small library of perfluoroaromatic linkers (Fig. 6c)[32]. It was envisioned that this automated system could be advantageous in the generation of stapled libraries from linear or disulfide-bridged hits from DNA-encoded libraries.

To execute this chemical transformation, additional reaction vessels and a magnetic hotplate stirrer were introduced to the base platform. Following assembly of the linear sequence, cleavage, and ether precipitation, the linear peptide was dissolved in DMF and transferred into four different reactors. Pre-made stock solutions of four perfluoroaromatic linkers (1 mM) and TRIS buffer (50 mM) in DMF were then added to each of the reactors containing the linear peptide solution via the liquid handling backbone. After stirring for 12 h, the crude reactions were then purified by RP-HPLC, affording milligram quantities of each of the perfluoroaromatic stapled peptides (**8-11**; 27–48% yield, 19 steps).

Having automated model one-step chemical modifications, we next tested the incorporation of multi-step chemical procedures into the automated system. The first multi-step chemical modification targeted was the synthesis and directed oxidative folding of Capitellacin (**12c**)[33]. Produced by the marine polychaeta *Capitella teleta*, Capitellacin adopts a β-hairpin conformation, stabilized by two disulfide bonds, and exhibits strong in vitro bactericidal activity against a wide panel of bacteria, including drug-resistant strains, making it a desirable target for potential medical applications[34,35]. To this end, the

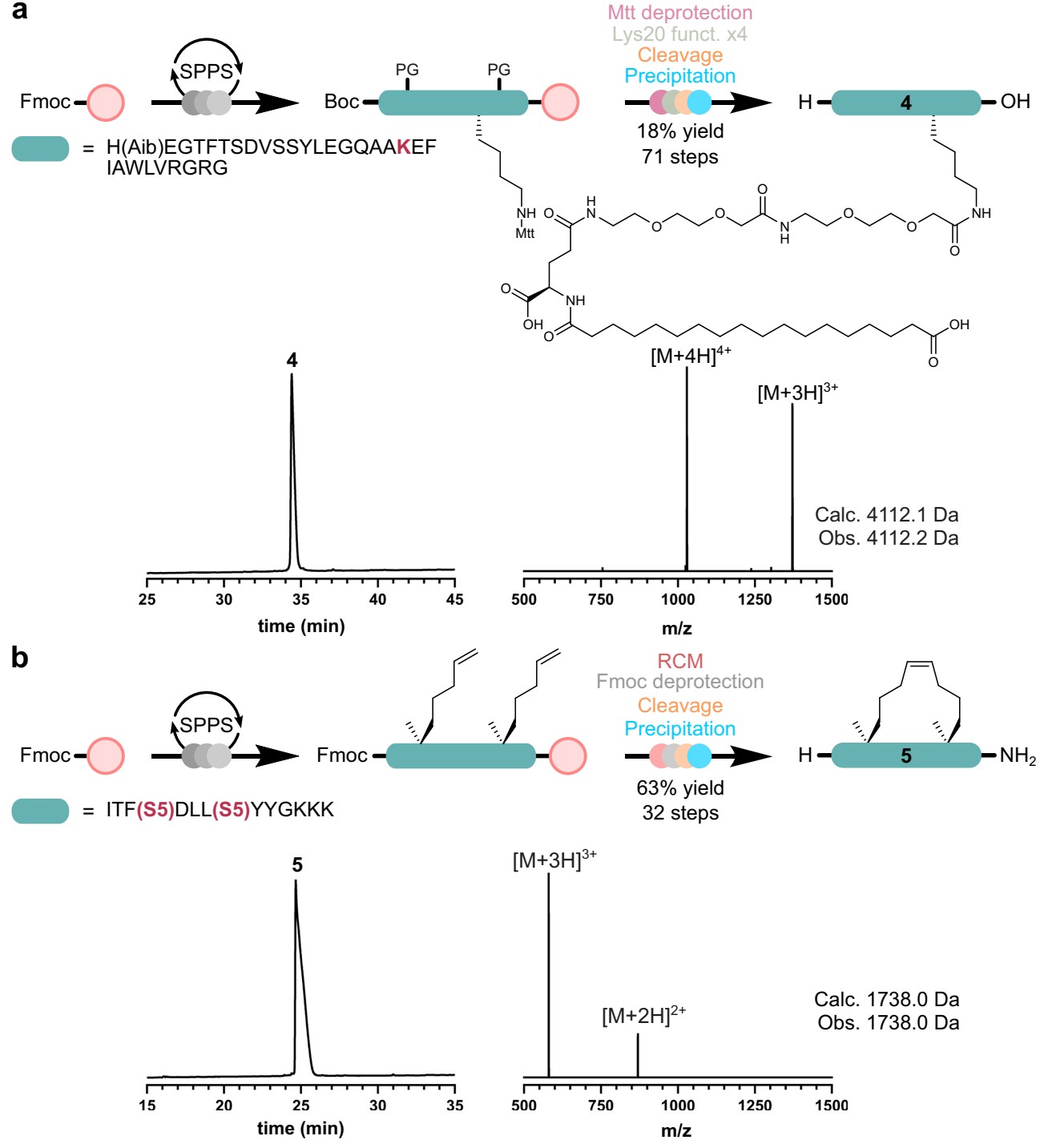

**Fig. 5 | Automated on-resin chemical modifications. a, b** RP-HPLC traces (214 nm, 60 min 0–80% MeCN gradient) and corresponding ESI-MS of the lysine fatty acid functionalized Semaglutide (**4**) and the all-hydrocarbon stapled NYAD-13 (**5**). All peptides were synthesized in full automation, including assembly, chemical modification, cleavage, and ether workup, and were purified by RP-HPLC. Mass spectra were obtained from the summation of the entire HPLC peak.

synthesis of this disulfide-rich natural product was selected to demonstrate the applicability of the SPPS Chemputer to give access to nature-derived bioactive compounds.

The synthesis of the correctly folded Capitellacin was facilitated by the directed oxidative folding of the two disulfide bridges at positions Cys5-Cys18 and Cys9-Cys14 (Fig. 7a). Synthetically, this was mediated by the inclusion of acetamidomethyl (Acm) protecting groups, inert to the standard cleavage conditions[36], at Cys5 and Cys18 during linear assembly. Following workup and precipitation, the Acm-protected peptide **12a** was dissolved in MeCN/$H_2O$ (1:9, v/v) and transferred to a reactor containing 260 mL of pH 8.3 ammonium bicarbonate buffer (0.1 M in $H_2O$) to oxidize the unprotected Cys residues (Cys2 and Cys8). After 48 h of stirring, the reaction was quenched with TFA to afford the partially oxidized peptide **12b**. A pre-made solution of iodine in glacial acetic acid (50 mM) was then added to the reactor vessel and stirred for 30 min, simultaneously removing the Acm protecting groups and oxidizing the newly exposed sulfhydryl groups to afford **12c**. The reaction was then quenched with ascorbic

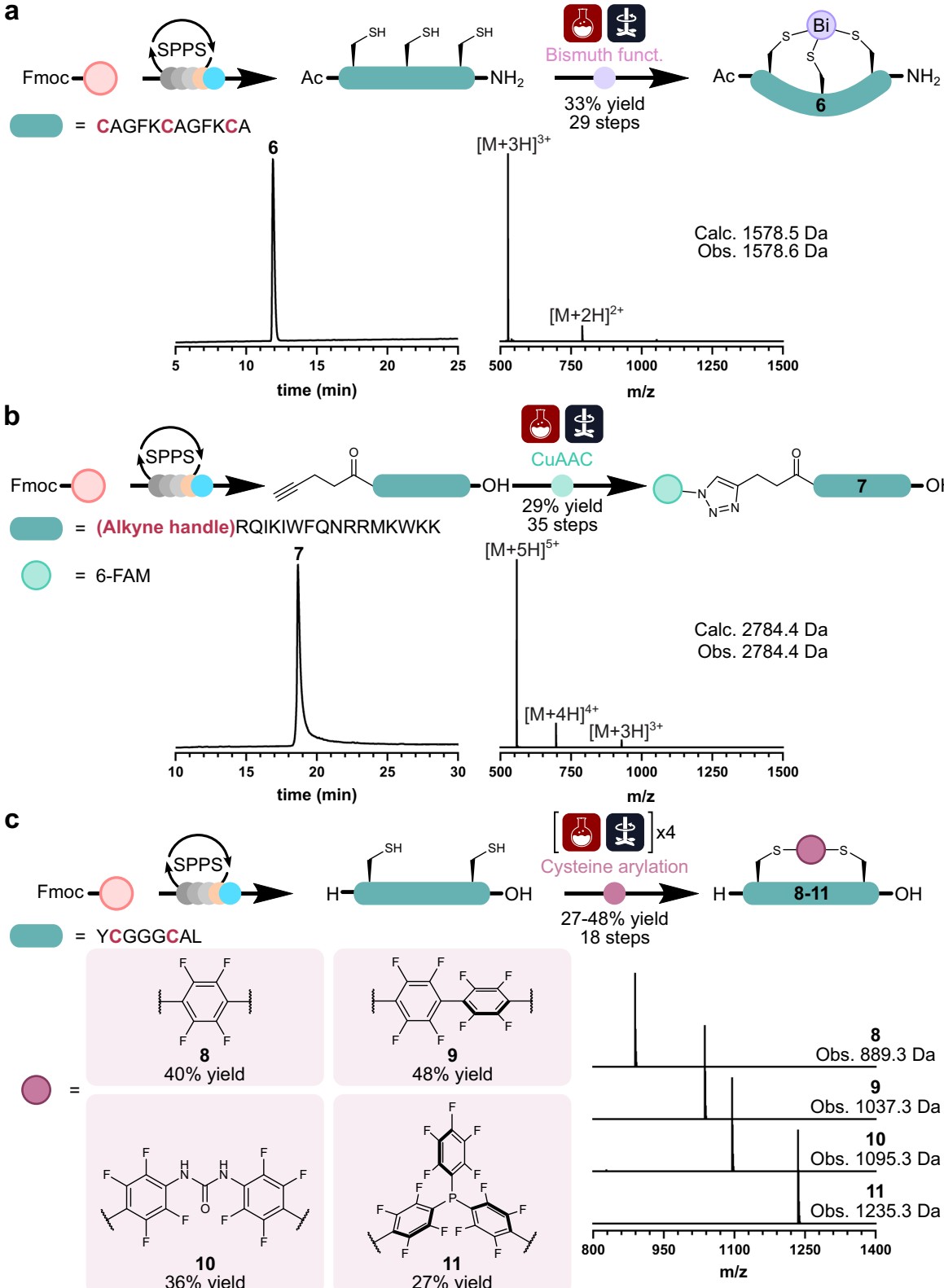

**Fig. 6 | Synthesis of chemically modified peptides in full automation. a, b** RP-HPLC traces (214 nm, 60 min 0–80% MeCN gradient) and corresponding ESI-MS of the automated synthesis of peptide–bismuth bicyclic complex (**6**) and CuAAC click functionalized cell penetrating peptide conjugated with a fluorescent tag (6-FAM) (**7**). **c** Late-stage peptide diversification via perfluoroaromatic cysteine arylation stapling (**8-11**) and corresponding ESI-MS. Purified RP-HPLC traces are shown in Supplementary Figs. 13–16.

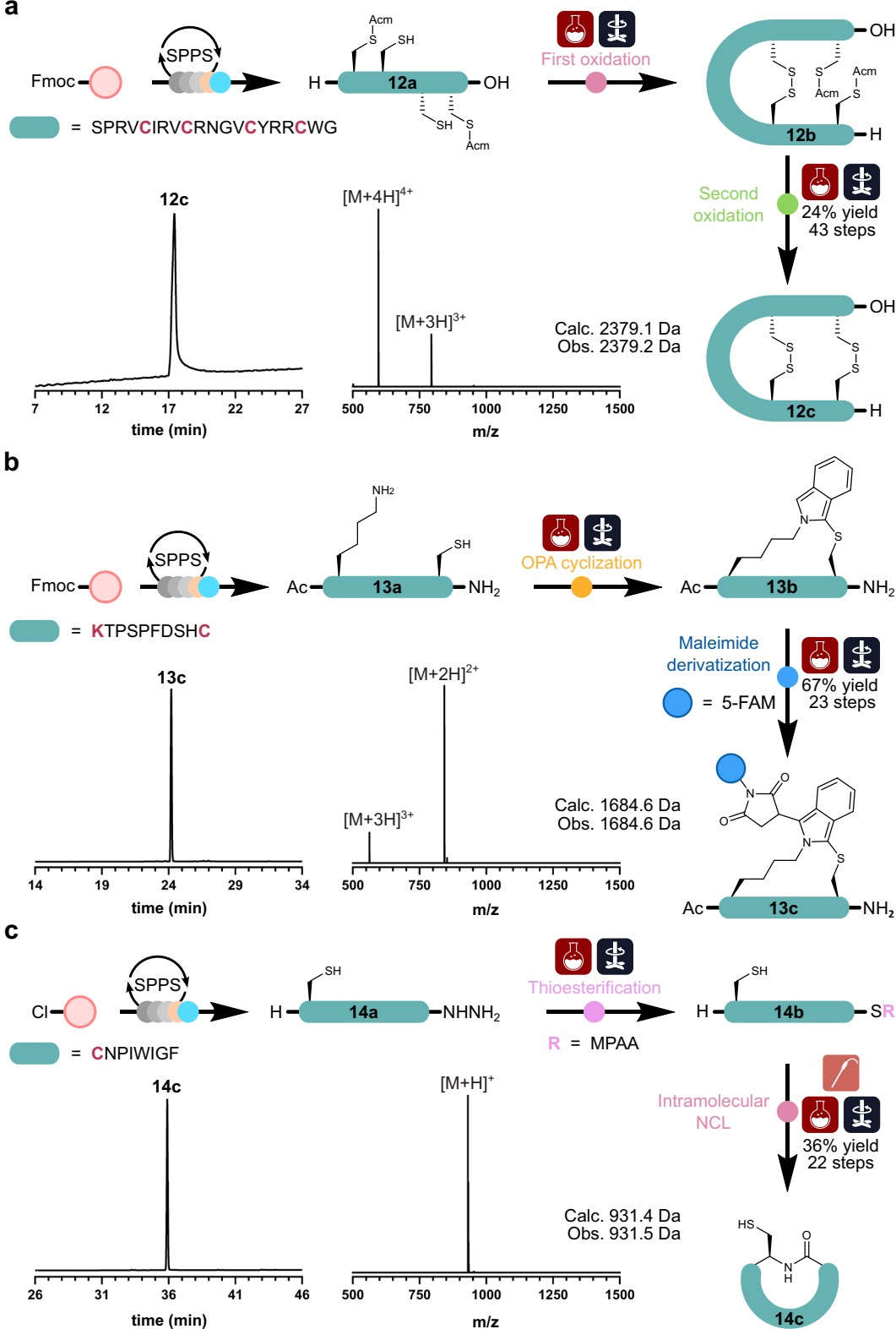

**Fig. 7 | Peptide synthesis and multi-step chemical modifications in full automation. a** Directed oxidative folding of Capitellacin (**12c**). **b** OPA-mediated side-chain cyclization and maleimide derivatization of a model peptide sequence (**13c**). **c** Thioesterification and intramolecular NCL of an α-amanitin peptide analogue (**14c**). RP-HPLC traces (214 nm, 60 min 0–80% MeCN gradient) and corresponding ESI-MS. Mass spectra were obtained from the summation of the HPLC peak at the time points depicted.

acid (1 M in H$_2$O) and purified by RP-HPLC to afford 37 mg of Capitellacin (**12c**; 24% yield, 43 steps) with its native disulfide connectivity.

The next multi-step reaction performed was the chemoselective peptide sidechain-to-sidechain cyclization reaction using bifunctional *ortho*-phthalaldehyde (OPA), followed by maleimide functionalization (Fig. 7b)[37]. Taking a model peptide sequence containing both a lysine and cysteine residue, peptide **13a** was assembled, cleaved, and precipitated in automation. The peptide precipitant was then dissolved in PBS buffer and transferred to a reactor. The unprotected peptide was then diluted to a concentration of 0.5 mM, before OPA (0.25 M in DMSO) was added dropwise over 15 min and stirred for another 30 min to afford **13b**. The reaction mixture was then treated with 5-maleimido-fluorescein (0.25 M in DMSO) and stirred for another 30 min to afford the final product **13c**. Following RP-HPLC, 60 mg of the isolated cyclic and functionalized peptide (**13c**; 67% yield, 23 steps) were obtained.

The final multi-step automation target chosen was NCL. First reported by Kent and co-workers in 1994, NCL revolutionized peptide synthesis, helping to overcome the limitation of standard SPPS[38]. NCL is a chemoselective process that enables the efficient and high-yielding ligation of two unprotected peptide segments—in aqueous solution and under mild reaction conditions—generating a native peptide bond. This ligation technique utilizes the unique reactivity of an N-terminal cysteine residue towards peptides equipped with a C-terminal thioester[39]. The study began with the synthesis of a model peptide **14a**, a non-toxic analogue of the α-amanitin toxin produced by Amanite mushrooms (Fig. 7c)[40]. The linear precursor was assembled using 2-chlorotrityl chloride (2-CTC) resin functionalized with Fmoc-hydrazide, prior to the first amino acid coupling[41]. Following assembly, cleavage, and precipitation, peptide **14a** was then dissolved in pH 3.0 guanidine hydrochloride buffer (Gn·HCl), with a final concentration of 3 mM, and transferred to the first reactor (Reactor 1) pre-charged with 4-mercaptophenylacetic acid (MPAA; 200 mM final concentration).

Following acetylacetone addition, to initiate the thioesterification reaction, the reaction mixture was stirred for 4 h to afford the α-thioester **14b**. The mixture was then transferred to a second reactor (Reactor 2) where **14b** was diluted to a final concentration of 0.5 mM with pH 7 NCL buffer—consisting of 6 M Gn·HCl, 0.2 M Na$_2$HPO$_4$, and 50 mM TCEP. The reaction pH was then adjusted using stock solutions of NaOH (1 M in H$_2$O) and HCl (1 M in H$_2$O) in complete automation using real-time sensor feedback. This was achieved using a pre-programmed analog pH sensor, an Arduino controller, and the χDL monitor step, which iteratively measures the pH of Reactor 2 and adds acid or base until the declared pH threshold is met[42]. Upon reaching the desired pH of 7.1, the contents of Reactor 2 were then transferred back to Reactor 1 and stirred for 12 h to afford the intramolecularly ligated peptide **14c**. The crude reaction mixture containing **14c** was then purified by RP-HPLC to afford 25 mg of the backbone cyclic product (**14c**; 36% yield, 22 steps).

In conclusion, from resin swelling to peptide workup, all steps of SPPS were captured using the Chemical Description Language χDL and automatically executed on the Chemputer. The platform demonstrated the ability to routinely produce difficult peptide sequences on micromolar scale, with high crude purities (>79%), in an uninterrupted workflow. Furthermore, high-value modifications were integrated into the automated process, affording on-resin and in-solution chemical derivatization of synthesized peptides. These include RCM, CuAAC, and intramolecular NCL. This unified platform finally unlocks a long-standing vision of digital chemistry, simplifying the synthesis of complex peptide sequences from commercially available starting materials, making this group of bioactive compounds accessible to a wider, non-specialized audience.

Looking ahead, we envision this technology as a critical enabler in closing the gap between organic chemistry and the biological sciences. The Chemputer platform's designs and codebase have been published online, and direct access is available through collaboration with the Cronin Lab, ensuring that this technology is accessible to the broader scientific community. By reducing the barriers traditionally associated with complex peptide synthesis, the Chemputer platform has the potential to democratize access to advanced biomolecules—eliminating gatekeeping and empowering a broader community of researchers.

## Methods
### General protocol for automated SPPS
Unless noted otherwise, all peptides were synthesized at a 0.1 mmol scale (based on resin loading) in complete automation, using the SPPS Chemputer following a standardized Fmoc-SPPS protocol. Solid resin supports used for SPPS were as follows: Fmoc-Gly-Wang (0.36 mmol/g), Fmoc-Rink Amide AM (0.42 mmol/g), Fmoc-Leu-Wang (0.56 mmol/g), Fmoc-Lys(Boc)-Wang (0.50 mmol/g), and 2-CTC (0.75 mmol/g). Stock solutions of Fmoc-protected amino acids and coupling reagents were freshly prepared before each synthesis using ultrapure peptide grade DMF (0.5 M and 0.45 M, respectively). All reactions were carried out at room temperature, unless stated otherwise. Following automated synthesis, peptide products were directly lyophilized or purified by preparative RP-HPLC in automation, following a manual transfer.

**Resin swelling.** Prior to peptide elongation, the resin solid support was first swelled in DMF (9 mL) for 1 h.

**Resin wash.** The resin was washed between each step using DMF (5 × 9 mL, 45 s).

**Synthesis of C-terminal hydrazide peptides.** Following swelling, 2-CTC resin was treated with a 0.5 M solution of Fmoc-hydrazide (2 mL, 1 mmol, 10 equiv. relative to resin loading) in *N*-methyl-2-pyrrolidone (NMP) (2 × 45 min). Peptide assembly was then performed in accordance with the standard Fmoc-SPPS protocol below.

**Deprotection.** Two-stage deprotection of the Fmoc protecting group was carried out using 20% piperidine solution in DMF (9 mL). The first portion of base was mixed with the resin for 3 min before draining. The second portion was then added and mixed for a further 12 min, for a total deprotection time of 15 min.

**Amino acid coupling.** Solution of protected amino acid (2 mL, 1 mmol, 10 equiv. relative to resin loading), coupling reagent (2 mL, 0.9 mmol, 9 equiv.), and DIPEA (0.5 mL, 2.9 mmol, 29 equiv.) were individually added to the resin, and the mixture sparged with nitrogen gas at room temperature for 30 min.

**N-terminal amine acetylation.** Following the final amino acid coupling and subsequent Fmoc deprotection, 9.5 mL of acetic anhydride (0.6 M in DMF) and 0.5 mL of DIPEA (final ratio acetic anhydride/DIPEA/DMF 1:2:17 v/v/v, 10 mL) were added to the SPPS reactor and the mixture sparged with nitrogen gas (2 × 3 min).

**Resin cleavage.** The resin was first washed with DCM (5 × 9 mL, 45 s) and then dried under vacuum (15 min). Then, a freshly prepared cleavage mixture containing TFA/H$_2$O/TIPS (90:5:5 v/v/v, 10 mL) was added to the resin and sparged with low-flow nitrogen. After 2 h, the TFA mixture was transferred to the Precipitating unit containing pre-chilled diethyl ether (180 mL, −20 °C), added in parallelized automation during the cleavage step. The resin was then washed with another 10 mL of TFA, which was also transferred to the Precipitating unit.

**Workup.** Peptides in the acidic cleaving solution were precipitated in pre-chilled diethyl ether (9:1 diethyl ether/cleavage mixture v/v) at −20 °C for 30 min. Following filtration, the peptide precipitate was

then washed with fresh, chilled diethyl ether ($3 \times 30$ mL). Finally, the solid peptide was heated to room temperature, dissolved in MeCN/$H_2O$ (1:1, v/v), and transferred to a receiving vial for lyophilization or dissolved in a suitable solvent and directly used for chemical modifications as described in the Supplementary Information.

## Data availability

All data are available in the manuscript, Supplementary Information, and from the corresponding author upon request.

## Code availability

The χDL, graph, and blueprint files generated in this study have been deposited in the Zenodo database under accession code 10.5281/zenodo.15948769. Further information is available from the corresponding author upon request.

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

## Acknowledgements

The authors would like to acknowledge the financial support provided by the EPSRC (grant nos. EP/S019472/1, EP/W001918/1, EP/R01308X/1, EP/S030603/1, and EP/S017046/1), NIH (project 1UG3TR004136-01), and The Eric And Wendy Schmidt Fund For Strategic Innovation (project G-21-62939). Furthermore, we thank the following people from the Cronin Laboratory at the University of Glasgow: R. Rauschen and N. Grocholski for helping implement the automated pH probe real-time analysis, and D. Thomas and C. Knittl-Frank for their assistance with manuscript proofreading.

## Author contributions

L.C. conceived the concept and the project idea, raised funds, and supervised the project. J.Z. and T.J.T. assembled the Chemputer platform, wrote synthetic χDL scripts, executed the syntheses, purified and analyzed products, and prepared the manuscript and Supplementary Information.

## Competing interests

The authors declare no competing interests.
