## [Transparent Peer Review file · Nature Communications]

Universal Peptide Synthesis via Solid-Phase Methods fused with Chemputation

Corresponding Author: Professor Leroy Cronin

Version 0:

Reviewer comments:

Reviewer #1

(Remarks to the Author)

This manuscript describes a fully automated, programmable platform for peptide synthesis merging traditional solid-phase peptide synthesis (SPPS) with the Chemputer. Using a digital chemical description language (χDL), the authors have digitized the entire SPPS workflow – from initial resin swelling and iterative amino acid couplings through cleavage and post-synthesis modifications – into a single automation protocol. The platform is demonstrated on fourteen peptides, including challenging long sequences and pharmaceutically relevant examples, as well as peptides requiring complex post-assembly modifications. Impressively, all these syntheses were achieved without manual intervention, yielding multi-milligram quantities of peptides with high crude purities and proper confirmation by HPLC and MS analyses. Overall, the work represents a significant technical advance in automated chemical synthesis. It addresses a long-standing bottleneck in peptide science – the inability of commercial peptide synthesizers to perform post-synthesis modifications or complex multistep reactions without human involvement. The manuscript is technically sound, very comprehensive, and generally well-presented. The innovation level is high, as to my knowledge this appears to be the first demonstration of a “universal” peptide synthesizer that can assemble peptides and carry out diverse chemical transformations. The findings will be of broad interest to chemists, chemical biologists, and the automation/robotics community. I have mostly positive impressions, with a few minor suggestions detailed below. In my opinion, after addressing these minor points, the manuscript will be suitable for publication.

(1) The authors may consider explicitly stating how reproducible each automated run is. If certain peptides were synthesized multiple times with similar outcomes, mention that to reinforce reliability.

(2) A short discussion about how this platform could be adopted by others would be welcome. For instance, are there plans to make the Chemputer setup available as a kit or via collaboration?

(3) While a full comparison study is not necessary, the manuscript could better articulate the advantages of the Chemputer approach in practice. For example, note that synthesizing a peptide like Semaglutide manually would require multiple separate stages (SPPS on a peptide synthesizer, then manual lipidation in solution, etc.), likely over several days with purifications in between, whereas here it is done in one go with minimal oversight. If any direct performance metrics are known (e.g., typical crude purity from a conventional synthesizer for similar length, or known difficulty of a sequence), mentioning those briefly would contextualize how well the automated system performs.

(4) In the concluding section, the authors could add a forward-looking statement, which would give readers a sense of how this technology might evolve and be applied next.

(5) Were there instances where automated protocols failed (e.g., clogged valves, precipitation issues), and how does the platform handle or recover from such errors? What is the rate of failure, if any, observed across multiple runs?

(6) Could the authors clarify whether the automated approach is practically scalable beyond the multi-milligram scale demonstrated here? Are there specific chemical transformations or steps that pose practical or chemical constraints at larger scales?

(7) What are the limits or restrictions regarding the range of solvents and reagents that the Chemputer system can handle?

Have the authors encountered any specific incompatibilities (e.g., corrosive reagents, highly viscous solutions)?

(8) Given that purification via HPLC was performed off-platform, is it technically feasible to integrate purification and in-line analytical characterization (e.g., LC-MS or NMR) directly within the automated workflow? If so, what practical steps or additional hardware would this integration require?

Reviewer #2

(Remarks to the Author)

I am not supporting the publication of the manuscript "Universal Peptide Synthesis via Solid-Phase Methods fused with Chemputation" on Nature Communications.

I read the manuscript, and for me, there is no novelty; the work describes one more peptide synthesizer, such as others in the market. I don't doubt that the Chemical Descriptive Language (χ DL) is well defined but is not the key point to obtain peptides of difficult sequence in high purity. In this regard the key point is the chemistry used, I mean, HATU/DIEA as the coupling reagent. I assume that the authors are not familiar with the peptide synthesis. HATU is probably the best coupling reagent, but it is not the first choice on both laboratory and industrial scales. The reasons are first, its high cost, and second, it is classified as potentially explosive. Additionally, the synthesis have been run using 10 times excess of reagents.

My opinion is that the manuscript is like manufacturer's brochure to sell a new instrument but with no interest for the Nature Communications readers.

Reviewer #3

(Remarks to the Author)

The manuscript titled "Universal Peptide Synthesis via Solid-Phase Methods fused with Chemputation," submitted by L. Cronin, reports the automated synthesis of 14 peptides using the Chemputer platform equipped with an χ DL-enabled SPPS module. Cronin and coworkers first introduced the Chemputer with χ DL in 2019 (ref. 12), and the submitted work utilizes this platform in combination with SPPS. The authors demonstrated the synthesis of 10- to 29-residue linear peptides (1–3) composed solely of proteinogenic amino acids, achieving high purity (>87%) with good yields (>40%). They also demonstrated on-resin chemical modifications, producing 14- to 31-residue bioactive linear and cyclic peptides (4 and 5) containing non-proteinogenic amino acids. Furthermore, they performed various post-assembly peptide modifications, including CuAAC click reaction, cysteine arylation stapling, sequential deprotection/S–S bond formation, sequential OPA cyclization/maleimide coupling, and sequential thioesterification/intramolecular NCL, resulting in peptides 6–14. The prepared peptides were manually injected into the RP-HPLC system for analysis and purification.

Automated SPPS was originally developed by Merrifield (ref. 7). In recent years, Pentelute and coworkers have reported automated continuous-flow SPPS for the ultra-rapid preparation of peptides and proteins. Universal frameworks capable of performing solution-phase multi-step synthesis have also been documented (e.g., review of automated synthesis of small molecules: *Angew. Chem. Int. Ed.* 2018, 57, 4192–4214). Among the most notable demonstrations are combinations of automated SPPS and multi-step solution-phase chemical modifications. This reviewer acknowledges that such combinations are indeed rare in previously reported automated synthesis. However, the submitted work does not appear to be significantly novel and gives the impression of being a study that skillfully combines existing technologies, including those previously reported by the authors themselves.

Although peptides with diverse structures are being automatically synthesized on a relatively large scale, and the optimization of the conditions described in the SI was intriguing, this reviewer finds it challenging to support the submission of this manuscript for publication in Nature Communications due to the insufficient significance and novelty of the work.

Additionally, this reviewer was concerned by the following relatively minor points:

The authors stated that "By translating literature procedures, chemical reactions typically performed manually by trained personnel can be digitally encoded and integrated into the automated SPPS workflow. Therefore, χ DL successfully integrates complex and highly specialized chemical steps into the automated operations routinely executed on commercially available peptide synthesizers. Any procedure from the chemical literature can then be directly translated into a χ DL script, fully encoding and digitizing not only SPPS but also the procedures required for post-assembly peptide modification."

While the translation and integration of literature procedures into the automated SPPS workflow are meaningful, it was unclear whether such translation and integration were actually carried out in the SPPS and multi-step solution-phase chemical modifications described in this manuscript.

The authors repeatedly refer to certain peptide sequences as "difficult" or "challenging" in the paper. However, the target peptides do not include components such as N-alkyl or N-aryl amino acids, which are typically difficult to couple. Thus, it remains unclear why the authors regarded these sequences as "difficult" or "challenging."

Version 1:

Reviewer comments:

Reviewer #1

(Remarks to the Author)

All my comments have been adequately addressed by the authors.

Reviewer #3

(Remarks to the Author)

After reading the pre-revision version of the manuscript, I was left with the impression that the study involved a skillful combination of existing technologies, including those previously developed by the authors. As such, I found it difficult to recommend publication at that stage.

However, upon reviewing the revised manuscript, I noted that the authors have responded constructively to the reviewers' comments and have convincingly demonstrated the efficient synthesis of peptides, including semaglutide, a blockbuster pharmaceutical and specialty peptide incorporating N-methyl amino acids—building blocks that are difficult to introduce via SPPS. This strengthens the case for the utility of the developed technologies. The revision also clarified, for this reviewer, the procedure for converting protocols from the literature into χ DL files.

Although I still find it difficult to assess the novelty of each individual component technology, the revision has increased the overall significance of the established combination of techniques. Consequently, I believe the value of reporting these developments has increased. Based on the content of the revised manuscript, I now consider it suitable for publication in Nature Communications.

REVIEWER COMMENTS: Universal Peptide Synthesis via Solid-Phase Methods fused with Chemputation

Referee reports in italics, our reply in normal type.

Reviewer #1 (Remarks to the Author):

This manuscript describes a fully automated, programmable platform for peptide synthesis merging traditional solid-phase peptide synthesis (SPPS) with the Chemputer. Using a digital chemical description language (χ DL), the authors have digitized the entire SPPS workflow – from initial resin swelling and iterative amino acid couplings through cleavage and post-synthesis modifications – into a single automation protocol. The platform is demonstrated on fourteen peptides, including challenging long sequences and pharmaceutically relevant examples, as well as peptides requiring complex post-assembly modifications. Impressively, all these syntheses were achieved without manual intervention, yielding multi-milligram quantities of peptides with high crude purities and proper confirmation by HPLC and MS analyses. Overall, the work represents a significant technical advance in automated chemical synthesis. It addresses a long-standing bottleneck in peptide science – the inability of commercial peptide synthesizers to perform post-synthesis modifications or complex multistep reactions without human involvement. The manuscript is technically sound, very comprehensive, and generally well-presented. The innovation level is high, as to my knowledge this appears to be the first demonstration of a “universal” peptide synthesizer that can assemble peptides and carry out diverse chemical transformations. The findings will be of broad interest to chemists, chemical biologists, and the automation/robotics community. I have mostly positive impressions, with a few minor suggestions detailed below. In my opinion, after addressing these minor points, the manuscript will be suitable for publication.

(1) The authors may consider explicitly stating how reproducible each automated run is. If certain peptides were synthesized multiple times with similar outcomes, mention that to reinforce reliability.

We have added the following sentence to page 8 of the manuscript, highlighting the high degree of platform fidelity presented in Supplementary Table 1:

“Notably, when testing different reaction conditions, a high level of reproducibility was observed, illustrating the platform’s reliability and consistency (Supplementary Table 1).”

(2) A short discussion about how this platform could be adopted by others would be welcome. For instance, are there plans to make the Chemputer setup available as a kit or via collaboration?

The Chemputer platform’s designs and codebase have been published online. Furthermore, we always welcome enthusiastic collaborators who seek to be early adopters in a rapidly advancing field. Alternatively, a Chemputer Platform Kit is currently in development, which will allow for facile deployment and seamless integration into the existing synthetic workflows of

other research laboratories. We are also currently raising funding to make an open initiative to help others build their own chemputers and expand the community.

We have also added the following sentence to page 17 of the manuscript:

“The Chemputer platform’s designs and codebase have been published online and direct access is available through collaboration with the Cronin Lab, ensuring that this technology is accessible to the broader scientific community.”

(3) While a full comparison study is not necessary, the manuscript could better articulate the advantages of the Chemputer approach in practice. For example, note that synthesizing a peptide like Semaglutide manually would require multiple separate stages (SPPS on a peptide synthesizer, then manual lipidation in solution, etc.), likely over several days with purifications in between, whereas here it is done in one go with minimal oversight. If any direct performance metrics are known (e.g., typical crude purity from a conventional synthesizer for similar length, or known difficulty of a sequence), mentioning those briefly would contextualize how well the automated system performs.

The authors agree with the sentiment of the reviewer and have added the following paragraph to page 9 of the manuscript:

“Following standard synthetic practices, this process would have been considerably more time-consuming and labor-intensive, requiring multiple manual steps and potential intermediate purifications, not to mention the historically manual cleavage and precipitation. In contrast, use of the SPPS Chemputer platform enabled the streamlined and automated synthesis of Semaglutide, resulting in a high crude purity of 57% (Supplementary Fig. 8a), exceeding previously reported values for the total chemical synthesis of the peptide (36–39%)¹. This highlights the capability of the Chemputer approach in accelerating complex peptide synthesis while improving both efficiency and product quality.”

1. Schönleber, R. O. & Loidl, G. *Solid Phase Synthesis of Acylated Peptides*. (Google Patents, 2021).

(4) In the concluding section, the authors could add a forward-looking statement, which would give readers a sense of how this technology might evolve and be applied next.

We have added the following concluding paragraph to page 17 of the manuscript:

“Looking ahead, we envision this technology as a critical enabler in closing the gap between organic chemistry and the biological sciences. The Chemputer platform’s designs and codebase have been published online and direct access is available through collaboration with the Cronin Lab, ensuring that this technology is accessible to the broader scientific community. By reducing the barriers traditionally associated with complex peptide synthesis, the Chemputer platform has the potential to democratize access to advanced biomolecules—eliminating gatekeeping and empowering a broader community of researchers.”

(5) Were there instances where automated protocols failed (e.g., clogged valves, precipitation issues), and how does the platform handle or recover from such errors? What is the rate of failure, if any, observed across multiple runs?

To mitigate potential mechanical failures, such as clogging of valves or precipitation issues, extensive cleaning protocols were incorporated into the synthetic workflow. Other instances of automated protocol failure (e.g., hardware disconnections) were quite rare and did not impede successful repetition of chemical syntheses. The Chemputer architecture can recognize when such events occur, pausing the synthetic execution and logging the issues observed. Ongoing efforts (for a future publication) are focused on improving the platforms flexibility, enabling more dynamic real-time monitoring, control, and recoverability of the Chemputer system.

(6) Could the authors clarify whether the automated approach is practically scalable beyond the multi-milligram scale demonstrated here? Are there specific chemical transformations or steps that pose practical or chemical constraints at larger scales?

With the current glassware setup, the practical upper limit for synthesis is at the gram scale (1–10 g). Even with the implementation of a larger solid-phase peptide synthesis (SPPS) reactor and an expanded precipitation unit, the process would likely remain limited to the low gram scale due to physical constraints associated with fritted glassware and solvent-handling capacity. To achieve higher material throughput, iterative and tandem synthetic runs could be employed, enabling access to hundreds of grams of material.

(7) What are the limits or restrictions regarding the range of solvents and reagents that the Chemputer system can handle? Have the authors encountered any specific incompatibilities (e.g., corrosive reagents, highly viscous solutions)?

While prolonged direct exposure to strong acids and bases can contribute to wear in the valves used within the Chemputer architecture, we did not observe any specific incompatibilities with the corrosive reagents commonly employed in our synthetic procedures, including trifluoroacetic acid (TFA), piperidine, dimethylformamide (DMF), and dichloromethane (DCM). Any degradation observed appeared consistent with expected long-term wear and tear, rather than acute chemical incompatibility. For solutions with particularly high or low viscosity, the Chemical Descriptive Language (χ DL) incorporates adaptive parameters that adjust the behavior of the Chemputer pumps to maintain accurate and reproducible reagent handling. This includes reducing the solvent transfer rate (e.g., from 40 mL/min to 5 mL/min) and introducing pauses at the apex of each pumping cycle to ensure complete uptake of the intended volume. These features allow the system to robustly manage a broad range of solvent and solution properties without compromising accuracy or reproducibility.

(8) Given that purification via HPLC was performed off-platform, is it technically feasible to integrate purification and in-line analytical characterization (e.g., LC-MS or NMR) directly within the automated workflow? If so, what practical steps or additional hardware would this integration require?

The integration of on/at-line analytical characterization tools—such as NMR, HPLC-DAD, and MS—into the Chemputer framework is feasible and proof-of-concepts have been demonstrated in previous work from our group^{2,3}. In this manuscript, we implemented real-time pH monitoring during the native chemical ligation (NCL) reaction as a representative case of on-line sensing; however, this could readily be substituted with several other analytical characterization methods, including but not limited to HPLC-DAD, MS, NMR, UV, color, temperature, pressure, and conductivity^{2,3}.

Regarding purification, while HPLC purification was performed off-line in the present study, automated purification workflows have previously been incorporated into the Chemputer platform—most notably via automated flash chromatography^{4,5}. Adapting the existing architecture for automation of preparative HPLC purification is feasible, primarily requiring the integration of compatible software and hardware components into the current modular system. With appropriate valve configurations and solvent-handling strategies, such an extension would allow for fully automated purification to be incorporated into future versions of the SPPS Chemputer platform.

2. M. Mehr, S. H., Caramelli, D. & Cronin, L. Digitizing chemical discovery with a Bayesian explorer for interpreting reactivity data. *Proc. Natl. Acad. Sci.* **120**, e2220045120 (2023).
3. Leonov, A. I. *et al.* An integrated self-optimizing programmable chemical synthesis and reaction engine. *Nat. Commun.* **15**, 1240 (2024).
4. Rauschen, R., Ayme, J.-F., Matysiak, B. M., Thomas, D. & Cronin, L. A programmable modular robot for the synthesis of molecular machines. *Chem* doi:10.1016/j.chempr.2025.102504.
5. Rohrbach, S. *et al.* Digitization and validation of a chemical synthesis literature database in the ChemPU. *Science* **377**, 172–180 (2022).

Reviewer #2 (Remarks to the Author):

I am not supporting the publication of the manuscript "Universal Peptide Synthesis via Solid-Phase Methods fused with Chemputation" on Nature Communications.

I read the manuscript, and for me, there is no novelty; the work describes one more peptide synthesizer, such as others in the market.

We respectfully disagree with the reviewer's assessment regarding the lack of novelty. This work represents the first demonstration of fully automated, end-to-end peptide synthesis—from resin swelling to peptide precipitation—executed entirely within the Chemical Descriptive Language (χ DL) framework.

Here we are able to **combine and unify** the ability to make organic molecules with χ DL with solid phase peptide synthesis. This unlocks the post-peptide programming of peptides, an achievement that allows us to extend robotic synthesis of complex-hybrid peptide based molecules that require additional organic reactions beyond the formation of peptide bonds.

This approach enables the synthesis of challenging peptide sequences with high purity on a multi-milligram scale, offering a level of integration and control not provided by existing technologies. Moreover, the high crude purity afforded by our automated workflow allowed us to carry out subsequent peptide modifications within the same uninterrupted protocol, further underscoring the versatility and power of the χ DL-based system. We believe these capabilities collectively distinguish our work and represent a meaningful advancement in automated synthesis.

I don't doubt that the Chemical Descriptive Language (χ DL) is well defined but is not the key point to obtain peptides of difficult sequence in high purity.

We strongly believe the robustness of XDL is central to achieving the synthesis of challenging peptide sequences in high purity. χ DL serves as the foundational architecture that enables the Chemputer—or any other automated synthesis platform integrated into the χ DL framework—to digitally capture, interpret, and execute each step of the peptide synthesis workflow, from resin swelling to peptide precipitation, in a reproducible and fully automated manner. This level of standardized digital control is critical for ensuring consistency, precision, and scalability across complex syntheses. It is precisely this framework that enables the reliable preparation of difficult peptide sequences with minimal manual intervention and high crude purity.

In this regard the key point is the chemistry used, I mean, HATU/DIEA as the coupling reagent. I assume that the authors are not familiar with the peptide synthesis. HATU is probably the best coupling reagent, but it is not the first choice on both laboratory and industrial scales.

We appreciate the reviewer's comment and agree that HATU is among the most effective coupling reagents available—which is precisely why it was selected for this study. However, we would like to clarify that the Chemputer platform is coupling reagent agnostic. This is demonstrated by our successful use of alternative reagents such as PyBOP and HBTU, which produced comparable results (see Supplementary Table 1). While we acknowledge that reagent selection may vary between laboratories and industrial settings, HATU remains a widely adopted choice in academic research due to its high efficiency and reliability⁶⁻⁸.

6. Rehm, F. B. H. *et al.* Site-Specific Sequential Protein Labeling Catalyzed by a Single Recombinant Ligase. *J. Am. Chem. Soc.* **141**, 17388–17393 (2019).
7. Hartrampf, N. *et al.* Synthesis of proteins by automated flow chemistry. *Science* **368**, 980–987 (2020).
8. Adams, Z. C. *et al.* Stretching Peptides to Generate Small Molecule β -Strand Mimics. *ACS Cent. Sci.* **9**, 648–656 (2023).

The reasons are first, its high cost, and second, it is classified as potentially explosive.

While we acknowledge the reviewer's concern regarding the use of HATU, we would like to reiterate that the Chemputer platform is coupling reagent independent. If cost is a limiting factor, alternative reagents can be substituted without significant loss of performance, as demonstrated in Supplementary Table 1. While it is true that HATU is classified as potentially

explosive, unfortunately this is also the case for many peptide coupling reagents—including PyBOP, HBTU, TDBTU, HDMA, TCTU, NDSC, TOTU, PyAOP, DMTMM, TNTU, and TPTU⁹. However, handling of such compounds is limited to the preparative stage only, leaving all other instances to the automated platform. Alternatively, more thermally stable coupling reagents could be used.

9. Sperry, J. B. *et al.* Thermal Stability Assessment of Peptide Coupling Reagents Commonly Used in Pharmaceutical Manufacturing. *Org. Process Res. Dev.* **22**, 1262–1275 (2018).

Additionally, the synthesis have been run using 10 times excess of reagents.

While it is true that an excess of coupling reagent is used in each step, this is standard practice in both manual and automated SPPS to ensure complete amide bond formation.

My opinion is that the manuscript is like manufacturer's brochure to sell a new instrument but with no interest for the Nature Communications readers.

We appreciate the reviewer's feedback; however, we respectfully disagree with the characterization of the manuscript as a promotional piece. The primary objective of this work is to present a fully automated platform that facilitates easy access to complex peptide sequences with late-stage multi-step chemical modifications. We believe this platform holds significant potential for a wide audience, as it dramatically lowers the barrier to entry and reduces the technical expertise required, thanks to the advantages of automation. Furthermore, we are confident that this work will foster further research and dialogue at the intersection of automation and chemistry, which aligns with the journal's focus on high-impact contributions.

Reviewer #3 (Remarks to the Author):

The manuscript titled “Universal Peptide Synthesis via Solid-Phase Methods fused with Chemputation,” submitted by L. Cronin, reports the automated synthesis of 14 peptides using the Chemputer platform equipped with an χ DL-enabled SPPS module. Cronin and coworkers first introduced the Chemputer with χ DL in 2019 (ref. 12), and the submitted work utilizes this platform in combination with SPPS. The authors demonstrated the synthesis of 10- to 29-residue linear peptides (1–3) composed solely of proteinogenic amino acids, achieving high purity (>87%) with good yields (>40%). They also demonstrated on-resin chemical modifications, producing 14- to 31-residue bioactive linear and cyclic peptides (4 and 5) containing non-proteinogenic amino acids. Furthermore, they performed various post-assembly peptide modifications, including CuAAC click reaction, cysteine arylation stapling, sequential deprotection/S–S bond formation, sequential OPA cyclization/maleimide coupling, and sequential thioesterification/intramolecular NCL, resulting in peptides 6–14. The prepared peptides were manually injected into the RP-HPLC system for analysis and purification. Automated SPPS was originally developed by Merrifield (ref. 7). In recent years, Pentelute and coworkers have reported automated continuous-flow SPPS for the ultra-rapid preparation of peptides and proteins. Universal frameworks capable of performing solution-phase multi-step

synthesis have also been documented (e.g., review of automated synthesis of small molecules: Angew. Chem. Int. Ed. 2018, 57, 4192–4214). Among the most notable demonstrations are combinations of automated SPPS and multi-step solution-phase chemical modifications. This reviewer acknowledges that such combinations are indeed rare in previously reported automated synthesis. However, the submitted work does not appear to be significantly novel and gives the impression of being a study that skillfully combines existing technologies, including those previously reported by the authors themselves.

We are unclear about the reviewer's concerns regarding the novelty of this work. To the best of our knowledge, this manuscript presents the first successful integration of peptide synthesis with a universal chemical framework, enabling peptide sequences to be synthesized and undergo seamless, multi-step solution-phase modifications within a single automated workflow. This integration represents a novel advancement in the field and significantly extends the capabilities of the Chemputer platform, originally introduced by our group in 2019¹⁰, empowered by the Chemical Descriptive Language (χ DL). We believe this work constitutes an important contribution to the scientific community by broadening access to peptide sequences that require late-stage, multi-step chemical modifications, thereby underscoring both the technical and conceptual innovation of the system.

10. Steiner, S. *et al.* Organic synthesis in a modular robotic system driven by a chemical programming language. *Science* **363**, eaav2211 (2019).

Although peptides with diverse structures are being automatically synthesized on a relatively large scale, and the optimization of the conditions described in the SI was intriguing, this reviewer finds it challenging to support the submission of this manuscript for publication in Nature Communications due to the insufficient significance and novelty of the work.

We appreciate the reviewer's feedback and acknowledge their concerns; however, we respectfully challenge their position. This work incorporates a broad range of chemical transformations, including on-resin and post-assembly in-solution modifications, executed within a unified system that surpasses current technological capabilities. By successfully integrating solid-phase peptide synthesis (SPPS) with multi-step solution-phase modifications into a seamless, fully automated workflow, we enable the synthesis of complex and diverse peptide sequences with high purity and precision. We believe this represents a significant and novel contribution, offering a scalable approach that facilitates the synthesis of biologically relevant peptides requiring intricate, multi-step chemical modifications, distinguishing our platform from existing technologies.

Additionally, this reviewer was concerned by the following relatively minor points: The authors stated that "By translating literature procedures, chemical reactions typically performed manually by trained personnel can be digitally encoded and integrated into the automated SPPS workflow. Therefore, χ DL successfully integrates complex and highly specialized chemical steps into the automated operations routinely executed on commercially available peptide synthesizers. Any procedure from the chemical literature can then be directly

translated into a χ DL script, fully encoding and digitizing not only SPPS but also the procedures required for post-assembly peptide modification.”

While the translation and integration of literature procedures into the automated SPPS workflow are meaningful, it was unclear whether such translation and integration were actually carried out in the SPPS and multi-step solution-phase chemical modifications described in this manuscript.

We are grateful for the reviewer’s feedback; however, we would like to clarify that all χ DL files used in this study are openly available in the supporting documentation, where the chemical steps executed by the Chemputer system are clearly outlined. To address the reviewer’s concerns more directly, we have revised the manuscript to explicitly reference the specific literature procedures that were translated into χ DL files and integrated into the automated workflow (please refer to the appropriate sections in the Supplementary Information 5.4–5.11). This revision should clarify the process by which these steps were incorporated into the automated SPPS and multi-step solution-phase chemical modifications discussed in the manuscript.

The authors repeatedly refer to certain peptide sequences as “difficult” or “challenging” in the paper. However, the target peptides do not include components such as N-alkyl or N-aryl amino acids, which are typically difficult to couple. Thus, it remains unclear why the authors regarded these sequences as “difficult” or “challenging.”

ACP(65–74) was selected as a model peptide since it has previously been described as “difficult” due to its propensity to aggregate during the deprotection and coupling of the final two residues^{11,12}. These interactions often lead to truncation events, making ACP(65–74) a valuable benchmark for evaluating synthetic platform performance¹³. To further assess the platform’s ability to handle peptides with a defined secondary structure, we chose 18A, known to adopt an amphipathic α -helical conformation¹⁴. Since our platform captures the entire SPPS workflow—including the critical precipitation step—we included 18A to evaluate the system’s ability to isolate peptides with secondary structures in automation. Finally, GHRH(1–29) was selected as a representative long-sequence, bioactive peptide. In addition to its biological relevance, GHRH(1–29) has been employed in previous studies as a benchmark for synthetic efficiency¹³. In response to the reviewer’s comment, we have revised the manuscript to include an additional peptide that incorporates N-alkyl amino acids (see Supporting Information 5.12 N-Methyl-18A (**15**)), which are widely recognized as particularly challenging for SPPS. This addition further strengthens our evaluation of the platform’s capability to handle synthetically demanding sequences.

11. Hancock, W. S., Prescott, D. J., Vagelos, P. R. & Marshall, G. R. Solvation of the Polymer Matrix. Source of Truncated and Deletion Sequences in Solid Phase Synthesis. *J. Org. Chem.* **38**, 774–781 (1973).

12. Zinieris, N., Zikos, C. & Ferderigos, N. Improved solid-phase peptide synthesis of 'difficult peptides' by altering the microenvironment of the developing sequence. *Tetrahedron Lett.* **47**, 6861–6864 (2006).
13. Simon, M. D. *et al.* Automated Fast Flow Peptide Synthesis. in *Total Chemical Synthesis of Proteins* 17–57 (WILEY-VCH GmbH, 2021).
14. Wolska, A., Reimund, M., Sviridov, D. O., Amar, M. J. & Remaley, A. T. Apolipoprotein Mimetic Peptides: Potential New Therapies for Cardiovascular Diseases. *Cells* **10**, 597 (2021).